# Germplasm Acquisition and Distribution by CGIAR Genebanks

**DOI:** 10.3390/plants9101296

**Published:** 2020-10-01

**Authors:** Michael Halewood, Nelissa Jamora, Isabel Lopez Noriega, Noelle L. Anglin, Peter Wenzl, Thomas Payne, Marie-Noelle Ndjiondjop, Luigi Guarino, P. Lava Kumar, Mariana Yazbek, Alice Muchugi, Vania Azevedo, Marimagne Tchamba, Chris S. Jones, Ramaiah Venuprasad, Nicolas Roux, Edwin Rojas, Charlotte Lusty

**Affiliations:** 1Alliance of Bioversity International and the International Center for Tropical Agriculture (Alliance of Bioversity and CIAT), Via dei Tre Denari 472/a, 00057 Maccarese (Fiumicino) Rome, Italy; i.lopez@cgiar.org (I.L.N.); p.wenzl@cgiar.org (P.W.); n.roux@cgiar.org (N.R.); 2Global Crop Diversity Trust (Crop Trust), Platz der Vereinten Nationen 7, 53113 Bonn, Germany; nelissa.jamora@croptrust.org (N.J.); luigi.guarino@croptrust.org (L.G.); charlotte.lusty@croptrust.org (C.L.); 3International Potato Center (CIP), Av. La Molina 1895, La Molina Apartado 1558, Lima 12, Peru; n.anglin@cgiar.org (N.L.A.); e.rojas@cgiar.org (E.R.); 4International Maize and Wheat Improvement Center (CIMMYT), Apdo. Postal 6-641, 06600 Mexico, D.F., Mexico; t.payne@cgiar.org; 5Africa Rice Center (AfricaRice), Station de M’bé, Bouaké, 01 BP 2511 Bouaké, Cote d’Ivoire; m.ndjiondjop@cgiar.org; 6International Institute for Tropical Agriculture (IITA), PMB 5320, Ibadan 200001, Oyo State, Nigeria; l.kumar@cgiar.org (P.L.K.); m.tchamba@cgiar.org (M.T.); 7International Center for Agricultural Research in the Dry Areas (ICARDA), P.O. Box 114/5055, Beirut, Lebanon; m.yazbek@cgiar.org; 8World Agroforestry (ICRAF), Box 30677, Nairobi 00100, Kenya; a.muchugi@cgiar.org; 9International Crops Research Institute for the Semi-Arid Tropics (ICRISAT), Patancheru 502 324, Telangana State, India; v.azevedo@cgiar.org; 10International Livestock Research Institute (ILRI), Box 30709, Nairobi 00100, Kenya; c.s.jones@cgiar.org; 11International Rice Research Institute (IRRI), Los Baños 4030, Laguna, Philippines; v.ramaiah@irri.org

**Keywords:** plant genetic resources for food and agriculture, genebanks, access and benefit sharing, multilateral system, CGIAR

## Abstract

The international collections of plant genetic resources for food and agriculture (PGRFA) hosted by 11 CGIAR Centers are important components of the United Nations Food and Agriculture Organization’s global system of conservation and use of PGRFA. They also play an important supportive role in realizing Target 2.5 of the Sustainable Development Goals. This paper analyzes CGIAR genebanks’ trends in acquiring and distributing PGRFA over the last 35 years, with a particular focus on the last decade. The paper highlights a number of factors influencing the Centers’ acquisition of new PGRFA to include in the international collections, including increased capacity to analyze gaps in those collections and precisely target new collecting missions, availability of financial resources, and the state of international and national access and benefit-sharing laws and phytosanitary regulations. Factors contributing to Centers’ distributions of PGRFA included the extent of accession-level information, users’ capacity to identify the materials they want, and policies. The genebanks’ rates of both acquisition and distribution increased over the last decade. The paper ends on a cautionary note concerning the potential of unresolved tensions regarding access and benefit sharing and digital genomic sequence information to undermine international cooperation to conserve and use PGRFA.

## 1. Introduction

Over the last three decades, under the auspices of the United Nations (UN) Food and Agriculture Organization (FAO), the international community has repeatedly committed itself to developing and maintaining a global system on plant genetic resources for food and agriculture (global system) (http://www.fao.org/agriculture/crops/thematic-sitemap/theme/seeds-pgr/gpa-old/gsystem/en/). This global system includes specialized international bodies that monitor the status of the conservation and use of plant genetic resources for food and agriculture (PGRFA), develop normative instruments when necessary, and support the implementation and use of those instruments. Furthermore, in 2015, the Sustainable Development Goals were adopted, including Target 2.5 concerning the sustainable management of genetic diversity, and the following target indicator focusing on PGRFA in particular: “[n]umber of plant genetic resources for food and agriculture secured in medium or long term conservation facilities” (http://www.fao.org/sustainable-development-goals/indicators/251a/en/)

Through their management of international PGRFA collections, the CGIAR Centers make important contributions to both the global system and SDG Target 2.5. Their contributions include assembling and conserving PGRFA, adding value to those materials through extensive characterization, evaluation, documentation, and health testing, and supplying samples that are free of quarantine pests and diseases to researchers, plant breeders, farmers, national and community genebanks, and seed companies around the world. The international collections hosted by the CGIAR Centers include over 760,000 accessions of crops, forages, and trees that were originally obtained from 207 countries, as well as pre-bred materials.

Over the last ten years, the CGIAR Centers’ genebanks have distributed more than 1.1 million PGRFA samples to recipients in 163 countries. (Data source: Online Reporting Tool (ORT), https://grants.croptrust.org). These transfers represent approximately 23% of all PGRFA transferred following the rules of the multilateral system of access and benefit sharing (multilateral system) created by the International Treaty on Plant Genetic Resources for Food and Agriculture (Plant Treaty). The multilateral system is the internationally sanctioned mechanism for PGRFA exchanges under the global system. The CGIAR breeding programs were the source of an additional 66% (approximately 3.3 million samples) of the PGRFA transferred through the multilateral system. The remaining 11% of materials exchanged through the multilateral system were transferred by organizations and individuals outside the CGIAR. (Source: Plant Treaty Secretariat).

Given their central position within the multilateral system, the CGIAR genebanks’ patterns of international acquisition and distribution of PGRFA over time are potentially significant proxies for the overall status and functioning of the global system in general, and institutions governing access to genetic resources and benefit sharing in particular. The CGIAR genebank managers previously participated in a study of factors affecting acquisitions by the CGIAR genebanks from 1984 to 2009 [1]. The study found that the following factors contributed to a significant drop in genebanks’ rate of PGRFA acquisitions from the mid-1990s to 2009: decreased levels of international support for collecting expeditions, overstretched staff, inability to characterize and evaluate the materials already collected, and challenges associated with targeting gaps in existing collections. It established that the most consistent overarching factor was “the highly politicized nature of access and benefit sharing issues at the international, national, and local levels, combined with low levels of legal certainty”. The study concluded on a millennial note, looking forward to the resolution of outstanding international tensions over access and benefit-sharing issues, and the full implementation of the multilateral system, with the result that more PGRFA would be made available to include in the international collections maintained by the CGIAR genebanks on behalf of the international community.

The research presented in this paper was initiated with the primary objective of revisiting the conclusions of the earlier study. We structured our research around the following questions: have the CGIAR genebanks’ rates of acquisition and distribution of germplasm changed in the last 10 years (2010–2019) as compared to the previous decade (2000–2009)? If there were significant changes, what factors—either external or internal to the CGIAR —contributed to those changes? Finally, recalling one of the main findings of the earlier study, how have international policy frameworks in particular affected the genebanks’ acquisitions and distributions of PGRFA? The data and methods we used to investigate these research questions, and our principle findings are set out and discussed below.

Before proceeding, we note that the earlier study which served as initial inspiration for the research presented here focused almost exclusively on CGIAR genebanks’ acquisitions of PGRFA [1]. In the early stages of our research planning, we decided to expand the scope of our investigation to also include the genebanks’ rates of germplasm distribution and contributing factors.

It is important to underscore that this paper focuses almost exclusively on acquisitions and distributions of PGRFA by the CGIAR Centers’ genebanks, and not by the Centers’ breeding programs (other than when the breeding programs access materials from, or donate them to, the genebanks). One reason for focusing on genebanks is that data concerning the genebanks acquisitions and distributions are easier to assemble as a result of the historical, CGIAR system-wide coordination of genebank activities. It would take considerably more time and resources to compile time-sequenced data for breeders’ acquisitions of germplasm in particular. Another reason for focusing exclusively on genebanks is that the mandates and contributions to the global system of the genebanks and breeders, while closely linked, are different and therefore amenable to separate studies. It is the responsibility of genebanks to assemble and maintain globally relevant collections of PGRFA, maintain the genetic integrity of conserved materials and make them available, in the form received by the Center, to recipients world-wide. The CGIAR breeding programs, on the other hand, develop new and improved materials which they distribute globally. Both make enormously important, but different, contributions to the global system. While it would be extremely interesting to examine the breeding programs’ experiences in this regard, it is beyond the scope of the research presented here.

## 2. Materials and Methods

The CGIAR Centers hosting international PGRFA collections are Africa Rice Center, International Center for Agricultural Research in the Dry Areas (ICARDA), International Crops Research Institute for the Semi-Arid Tropics (ICRISAT), International Institute for Tropical Agriculture (IITA), International Potato Centers (CIP), International Rice Research Institute (IRRI), International Livestock Research Institute (ILRI), Alliance of Bioversity International and the International Center for Tropical Agriculture (Alliance of Bioversity and CIAT), International Maize and Wheat Improvement Center (CIMMYT), and the World Agroforestry (ICRAF). To assemble data on the genebanks’ acquisitions and distributions from 2013 to 2019 inclusive, records were compiled from the CGIAR Genebank Research Program 2012–2016 and the CGIAR Genebank Platform 2017–2021. These records are stored on the Online Reporting Tool (ORT) maintained by the Global Crop Diversity Trust (Crop Trust), coordinator of the CGIAR Genebank Platform. The genebanks’ managers reviewed those records, and provided additional information, subdividing acquisitions from each year into three mutually exclusive categories: from Centers’ own breeding programs, from new collecting expeditions, and donations from other organizations (from outside the CGIAR) Additional information regarding the countries and crops from which Centers acquired PGRFA was assembled through reference to Centers’ collecting expedition reports in the ORT and each genebank’s own records. For both acquisitions and distributions by the genebanks for 2010–2012 inclusive, each genebank provided the requisite annual acquisition and distribution data, including the subdivision of acquired materials into three categories of sources, provider countries, and crops.

Each Center genebank also provided aggregate totals of their annual acquisitions and distributions for 2005–2009 inclusive. While some of these data had been compiled for CGIAR reports to the Plant Treaty’s Governing Body and the FAO Commission on Genetic Resources for Food and Agriculture (CGRFA), this had not been done according to calendar years, so could not be used for this study. These data were combined with pre-existing data on annual acquisitions and distributions by the Centers covering the period of 1980–2004 (from the 2012 study referenced above).

Genebank managers were also asked to respond to a survey (see Appendix A) providing reflections on trends in acquisitions and distributions from 2010 to 2019, contributing factors, and experiences acquiring (or attempting to acquire) new materials during the last five years. Staff from the Centers’ genebanks, the Crop Trust, and the Genebank Platform Policy Module subsequently worked together in group teleconferences to review and synthesize key findings from the assembled data and survey.

The respondents/informants provided expert knowledge concerning genebanks’ performance targets and quality management standards, methods for identifying and prioritizing gaps in collections to be addressed through new collecting expeditions, Centers’ efforts to ensure healthy, quarantine organism-free genetic materials, and international crop conservation strategies. Literature reviews were conducted to gain insights into the experiences of genebanks outside the CGIAR. To validate data and key findings, several consultations with CGIAR genebank managers and staff were conducted.

## 3. Findings

### 3.1. Acquisitions

There was a dramatic rise in acquisitions of new PGRFA by the genebanks between 2010 and 2018, compared to the previous 10 years, though still lower than peaks reached in the 1980s and 1990s, as illustrated in Figure 1 and Figure 2 below. The increase over the last 10 years reached its height in 2012, when the genebanks received almost 14,000 samples of distinct PGRFA to include as new accessions in the international collections. Of course, not all of the materials that the genebanks receive are ultimately accessioned. If materials received are not viable, or are redundant with respect to the materials already in the collection, they are not included in the collections. For the convenience of using a short form, the tables and figures presented below make reference to “accessions”; in fact, the data refer to materials received with the intention of including them as accessions, assuming they are viable and not redundant. In 2019, the number of newly acquired materials by the Centers to include in the international collections dropped back down to the lower levels that characterized the mid-1990s to 2009.

In total, over the course of ten years, from 2010 to 2019 inclusive, the CGIAR genebanks acquired 116,921 samples of distinct PGRFA to include in their Article 15 collections.

Approximately 65% of the materials acquired by the genebanks came from providers in 142 different countries. The remaining 35% came from the Centers’ own breeding programs, as discussed below. A total of 84% of those countries are developing countries or countries with economies in transition as defined in the International Monetary Fund’s World Economic Outlook Database (October 2018, accessible at: https://www.imf.org/external/pubs/ft/weo/2018/02/weodata/groups.htm). Approximately 18% of the materials from countries came from new collecting expeditions; the other 82% was material that was already in ex situ conditions prior to being sent to the Centers.

All of the materials from providers in countries were either received under the standard material transfer agreement (SMTA) adopted for exchanges of materials under the Plant Treaty’s multilateral system (which allows the Center to conserve, use, and pass on the materials with the same SMTA) or under other agreements whereby the providers gave the Centers permission to subsequently distribute the material concerned with the SMTA. In this context, it is interesting to note that 31 of the countries from which the materials were made available are not currently contracting parties to the Plant Treaty, yet they were still willing to provide the materials, and have them subsequently redistributed by the CGIAR Centers, under the conditions established by the Plant Treaty. We did not analyzed whether some other countries which are now Plant Treaty members made materials available before becoming members.

Approximately one third of the PGRFA acquired by the genebanks from providers in countries between 2010 and 2019 were associated with a project coordinated by the Crop Trust from 2007 to 2012 called “Securing the Biological Basis of Agriculture” (hereinafter referred to as the “Regeneration project”), funded by the Bill and Melinda Gates Foundation and the Grains Research and Development Corporation. That project provided financial and technical support for organizations around the world to regenerate unique ex situ PGRFA that were at risk of being lost, to send a copy of the regenerated materials to an internationally recognized genebank, to send a copy for safety back-up in the Svalbard Global Seed Vault, and to make the materials available through the multilateral system of access and benefit sharing. Activities with 84 national partners in 54 countries resulted in the regeneration of approximately 73,000 threatened accessions, of which more than half were duplicated in CGIAR genebanks with permission to make them available through the multilateral system.

As of 2019, another 2256 samples of 1508 unique accessions collected from 25 countries were sent to CGIAR Center genebanks (ICARDA, ICRISAT, IRRI, IITA, CIP) by the Millennium Seed Bank (MSB), associated with the project called “Adapting Agriculture to Climate Change: Collecting, Protecting and Preparing Crop Wild Relatives” (hereinafter the “CWR project”) coordinated by the Crop Trust from 2011 to 2021 [2]. The CWR project, with funding from the Norwegian government, provided financial and technical support for project partners to target and collect wild species related to crops, to create a safety back-up, and make collected material available through the multilateral system. It is critically important to have safety duplication for PGRFA accessions hosted by organizations that can ensure the required conditions for long-term storage. Details concerning the Centers, crops, providing countries, and related programs are set out in Table 1 below.

As mentioned above, approximately 35% of the materials acquired by the genebanks between 2010 and 2019 came from the Centers’ own plant breeding programs, mostly from the CIMMYT wheat breeding program. Every genebank has a policy and process, both of which are periodically reviewed, for strategically acquiring materials from breeders’ collections for incorporation into the genebank for long-term conservation with the express aim of ensuring that incoming materials are likely either to represent a highly demanded material or diversity that is not already contained in the collection.

The availability of funds—both for providers and for the Centers as recipients—was one of the most frequently mentioned factors by the genebank managers affecting the ability of CGIAR Centers to acquire new materials to include in the genebanks.

The Centers representatives confirm that in many of the instances where the genebanks were able to acquire new materials, it was critically important to be able to provide financial and technical support for providers’ activities related to the collection, regeneration, phytosanitary cleaning and inspection, and shipping of accessions. In some cases, the Centers provided this support, in other instances—most notably, the Regeneration and CWR projects—the support came from other organizations.

Generally speaking, the financial costs of preparing and sending materials are not particularly high. Costs arise because of the lack of capacity and means to multiply, dry, test, and clean seed that is either collected or conserved and is only available in small quantities with unknown viability. The transaction costs are greatly higher when vegetatively propagated germplasms, including roots, tubers, bananas, or trees, are involved because the movement across international borders involves stringent phytosanitary restrictions that demand expensive disease cleaning and testing upon both shipping and receipt along with extensive periods of time in quarantine based on the risk assessed by the National Plant Protection Organizations (NPPO). In such cases, the Centers need to make substantial investments in strengthening partners’ capacity to test and clean materials.

Once materials arrive at the Center, they need to be processed through post-entry quarantine, health tested, and cleaned of infectious diseases, multiplied, tested for viability, and dried before being introduced into the collection and made available for distribution. In the case of clonally propagated crops, the steps can be very expensive and time-consuming, taking generally four years or longer for a new accession to become available (N. L. Anglin pers comm). Post-entry quarantine (PEQ) procedures such as growing the first generation in a quarantine greenhouse—a requirement for all newly acquired materials by the genebanks—are also expensive. Occasionally, the NPPO requires new acquisitions to remain in PEQ from four months up to two years to assess risks.

Direct costs for new collecting missions are also relatively modest. Of course, the costs per samples of materials collected greatly differs by crop type, wild or cultivated form, and geographical location. For example, similar projects, with similar budgets, working with national agricultural research system (NARS) partners to conduct collecting missions resulted in very different numbers of acquisitions; samples of 307 landraces and 94 wild relatives and forages were gathered in Tajikistan and Lebanon for the same cost as samples of 106 bananas collected in Papua New Guinea, Samoa, and the Cook Islands. The per accession costs will differ between seed and clonal crop collections by an order of magnitude more when taking into account the costs of incorporating the materials into the collections.

In addition to direct costs associated with the management and transfer of biological materials, there are transaction costs associated with getting requisite permissions to provide/access materials from both ex situ and in situ conditions. Transactional costs associated with these activities are often substantially increased in situations where the national policies and laws are unclear, or non-existent. In such cases, it can take extended rounds of communications over long periods of time with many different levels of national R&D partners, lawyers, and competent authorities before final decisions to provide materials can be made, or, as sometimes occurs, no final determination is ever communicated. Unlike some national genebanks or networks of collections [3], CGIAR does not have a centralized service that takes responsibility to help Centers comply with requisite processes for organizing collecting missions/partnerships, and obtaining requisite permissions.

Indeed, along with availability of funds, the CGIAR genebank managers emphasized that “restrictive or unclear laws or policies” were leading variables influencing their ability to acquire new PGRFA to include in the international collections. They note that the Plant Treaty’s multilateral system has contributed stability and a sound legal basis for providing and receiving germplasm, and that their ability to acquire materials through new collecting is evidence of cooperation/coordination between national authorities responsible for implementing the Plant Treaty and those responsible for regulating access to genetic resources outside the multilateral system (including those implementing the CBD or Nagoya Protocol). However, the Centers are concerned that unresolved disputes concerning the enhancement of the Plant Treaty’s multilateral system of access and benefit sharing, and digital sequence information (DSI) in particular (both under the Plant Treaty and the Nagoya Protocol), are holding back some countries (and providers within countries) from making more PGRFA available through the multilateral system. If international tensions over these issues remain unresolved for too long, they could further undermine the Centers’ ability to access, generate, use, and distribute PGRFA and associated information.

The Center genebanks confirm that, over time, they tend to obtain materials to include in their collections from the same set of countries or subregions, where they have established connections. By corollary, there are some countries and subregions from which they rarely, if ever, obtain materials. A number of the genebanks confirm that they rarely make overtures to organizations and or countries which have strongly signaled in the past that they are unwilling to make new materials available. Their general perception is that, despite the coming into force of the Nagoya Protocol and the existence of the Plant Treaty’s multilateral system of access and benefit sharing, some of these same countries have not substantially altered their approach to making materials available upon request for inclusion in the international genebanks’ collections.

The initial stimulus, and subsequent financial and technical support for PGRFA providers, from these internationally coordinated projects were clearly critically important factors contributing to the extraordinary increase in PGRFA that was made available for the CGIAR Centers to include in the international collections and thus the multilateral system.

The Centers genebanks’ representatives also report that the strength of the Centers’ long-term relationships with providers and provider countries is equally important. Most materials are made available to Centers as part of projects with providers. Centers rarely obtain new materials to include in the genebank as a result of “cold-calling” would-be providers with simple requests for materials and no other form of engagement. Out of appreciation of these factors, the Centers most recent collective efforts under the CGIAR Genebank Platform (suspended temporarily due to the COVID-19 pandemic) to catalyze and support new collecting expeditions involve support for “two-way flows” of germplasm from the genebanks to the providers, identified on the basis of a jointly conducted analysis of potentially useful germplasm to respond to local needs, and from the providers to the genebanks, and financial and technical support for institutional capacity building for partner organizations in the country concerned. This is consistent with practices of other organizations seeking to acquire PGRFA to include in public genebanks through new collecting activities [4].

Of course, these factors affecting the ability of the genebanks to acquire new materials need to be considered within the broader context of what additional PGRFA needs to be collected, backed up, and conserved as part of the global system, either by CGIAR genebanks or other organizations hosting globally available PGRFA collections. Methods for conducting gap analyses and strategies for coordination with other organizations are considered below.

From 2007 to 2017, CGIAR Centers acquired PGRFA through collecting expeditions from 14 countries. Collecting in six of those countries was supported by the CWR project. During the same period of time, the Center for Genetic Resources in the Netherlands (CGN, with 23,000 accessions) received materials through collecting expeditions in five countries, the Leibniz Institute of Plant Genetics and Crop Plant Research (IPK, with 105,000 accessions) received materials from collecting in six countries, and the National Plant Germplasm System of the United States Department of Agriculture (NPGS-USDA, with 500,000 accessions) organized collecting expeditions in at least 20 foreign countries [5,6,7,8,9,10,11]. Accessions from collecting expeditions in other countries account for 8% of CGN total germplasm acquisitions [12]. At the beginning of the century, such materials represented 20% of material acquired by NPGS-USDA [3]. In the last decade, collecting expeditions have contributed 11% of all the acquisitions by CGIAR genebanks. The fact that, relative to the cumulative size of their collections, the CGIAR Centers have engaged in relatively fewer collecting activities during this period than these other organizations may be attributable to a combination of the following factors. The diversity of some CGIAR mandate crops is relatively well represented in ex situ collections when compared to the crops that IPK, CGN, and NPGS-USDA have been prioritizing in their collecting activities, i.e., vegetables (e.g., lettuce, Allium, Brassica, chicory, spinach, asparagus, and carrot), fruit and nut trees (e.g., apple, pear, pomegranate, pistachio, walnut, and hazelnut), berries (Fragaria, Rubus, and Ribes), and temperate grasses (Poa, Festuca, Agrostis, Koeleria, and Puccinellia). IPK, CGN, and NPGS-USDA have been acquiring most of their new materials from the Transcaucasia and Central Asia regions and Europe [10,11], which are arguably more open to allowing new collecting missions than other countries and regions in the world. IPK, CGN, and NPGS-USDA may also have had more, and more regular, financial resources to dedicate to supporting new collecting activities. We acknowledge that we are only scratching the surface of potential comparisons between CGIAR and other genebanks around the world; we hope to be able to deepen such analyses in the future.

### 3.2. Distributions

Over the course of the last 10 years, the CGIAR genebanks have distributed on average 115,000 samples of germplasm per year around the world. While there are significant fluctuations in Centers’ distributions from year to year, the overall rate of Centers’ distribution from 2010 to 2019 is higher than the previous decade at 95,000 samples per year (2000 to 2009 annual average), as portrayed in Figure 1 above. Comparing the total distributions between time periods 2000–2014 and 2015–2019, four Centers increased their annual distributions in the latter half of the last decade, four remained generally the same, and three decreased.

There is considerable fluctuation, from year to year, in the ratio of materials the CGIAR genebanks send to recipients within the CGIAR (mainly breeders) and to recipients outside the CGIAR, as can be seen in Figure 3. Since 2017, the Centers genebanks have been distributing proportionately more materials to recipients outside the CGIAR. Some Centers do not have crop breeding programs (e.g., Bioversity, ILRI), so almost all of their distributions are to recipients outside the CGIAR. In 2019, the Centers and crops with a high proportion of materials distributed to breeders (within their own Center or in other Centers within the CGIAR) are AfricaRice (rice); ICRISAT (pigeon pea, chickpea); ICARDA (grasspea, barley); CIP (sweet potato); IRRI (rice); IITA (cassava); and CIMMYT (wheat).

Between 2017 and 2019, approximately 80% of PGRFA samples distributed by the genebanks to recipients outside the CGIAR were to a combination of NARS partners, national genebanks, advanced research institutes (ARIs), and universities (see Figure 4 for more details). The proportion of samples distributed to farmers, farmer organizations, and non-governmental organizations (NGOs) during this time is still relatively small (8%), approximately the same rate as noted earlier for 2015 [13]. CIP and ICRAF were the two centers whose proportionate distribution of materials to farmers and NGOs was largest, between 2017 and 2019 inclusive, as seen in Figure 4b. Much of the material distributed by CIP during this period was part of their repatriation program in which desired potato landraces are matched to farmers’ descriptions and returned to the farmers that have lost them due to normal attrition, environmental impacts (drought, hail, diseases), or other reasons, to help maintain and/or increase their diversity in farms. CIP also gives material to farmers in the repatriation program to support their needs in responding to climate change or to help improve yield.

In terms of absolute numbers, the largest provider of germplasm materials was CIMMYT, distributing 28% of the materials cumulatively distributed by the genebanks between 2010 and 2019, followed by IRRI and ICRISAT, distributing approximately 26% and 12%, respectively.

Landraces are the most frequently requested materials (50% between 2017 and 2019), followed by breeding materials (24%), and wild relatives (13%) (see Figure 5 for more details).

The countries which received the highest numbers of samples from CGIAR genebanks, between 2017 and 2019, are set out in Table 2. These data do not include transfers within or between CCIAR Centers. Four of these countries are not contracting parties to the Plant Treaty.

The genebanks’ annual distribution is a reflection of demand from users, often driven by the needs of the research and development projects in which they are involved, and by the profile or visibility of the genebanks among research and development, and non-governmental organizations in different countries. The Center hosting the genebank is frequently involved in the research in some way. Here, we provide some examples of projects with which distributed material was associated. The overall spike in distributions in 2013 is largely attributable to the internal CIMMYT transfers of 36,500 wheat accessions as part of CIMMYT’s Seeds of Discovery (SeeD) project to CIMMYT researchers and breeders characterizing maize and wheat genetic diversity for use in breeding programs. CIMMYT’s high number of distributions in 2013 also includes 8500 wheat accessions to recipients in Iran and Turkey as part of those countries’ restoration efforts. In 2019, ICRAF distributed considerably more tree germplasm than in recent years as a result of requests from organizations involved in the mega project “Regreening Africa” that is being implemented in East and West Africa. Between 2017 and 2019, the ICARDA genebank’s rate of distribution to recipients outside the CGIAR increased 200–300% over the previous five-year period, reflecting a combination of the Center’s progress getting its genebank’s operations back “up to speed” after the disruptions associated with having to relocate from Syria and a surge of new interest in the genebank’s materials as a result of publicity generated when the Center retrieved materials from the Svalbard Global Seed Vault. A big increase in the ICRISAT genebank’s distributions in 2014 and 2018 was associated with supporting a consortium of Indian organizations conducting chickpea and pigeon pea multilocation trials. In 2017, ICRISAT genebank’s distributions were higher than usual in part because it responded to a request from Italian organizations for core collections of cereals (sorghum, pearl millet, finger, kodo, proso, barnyard, little, and foxtail millets) as part of a program to reproduce local varieties and develop new crops adaptable to Italian local conditions. The IITA genebank’s increased rate of distributions in 2015 was associated with the requests for cowpea and Bambara groundnut germplasm from Nigerian universities to support their research programs on those crops.

The incidence of a newly emerging rare disease can also lead to a need to screen a large number of accessions for resistance [14,15].

The genebank managers confirm that one of the most important factors affecting demand for PGRFA is the quality and relevance of the accession-level information that the Centers compile about the materials in their collections, a finding consistent with the relevant literature [16,17]. Accession-level information helps users make informed decisions about what materials from the collections are potentially most useful for their specific purposes. It also makes it possible for the genebank to make more targeted selections of materials in response to their requests since often users do not know where to begin in choosing an appropriate accession for their needs. The genebanks highlighted the importance of trait-specific data including nutritional qualities, biotic and abiotic stresses, agronomic performance, genetic sequence information linked to traits or to provide information about relationships among accessions, and geographic information about place of collection, including climate conditions and soil type. IRRI reports there was a significant increase in the number of requests for genetic stocks of 3000 rice accessions, particularly between 2015 and 2018, after the full genome sequences were made publicly available. The genebanks have minted digital object identifiers under the Plant Treaty’s global information system (GLIS-DOIs) for almost all of the accessions in the Center-hosted international collections, with a long-term view towards helping this process and linking publications and data back to accessions.

While such information is absolutely necessary to generate interest in, and demand for, materials in the genebanks, it also permits users to make more targeted requests for a narrower range of materials with each request. Otherwise, users must consider thousands of accessions from which to choose. Thus, documenting the traits and uniqueness of each accession helps create a path for utilization of the germplasm. Further, creating cores or subsets of accessions, where small groups of accessions are defined by the genebank to help narrow down the search for specific traits, has had an influence on demand and could be scaled up significantly. AfricaRice’s genebank has used molecular markers to create core and mini-core sets that represent the maximum possible genetic variation contained in the African rice whole collection [18]. As a result, there has been a significant increase in requests for the mini-core set for use in rice genetic and breeding studies, and gene discovery (M.N. Ndjiondjop pers.com). The Centers’ genebank managers note that as they have been able to increase the quantity of accession-level information, requests are indeed becoming more informed and better targeted. Scientists working with genebanks both outside [19,20] and with CGIAR are developing methods to assist users to identify useful materials from ex situ collections, including the ICARDA-led focused identification of germplasm strategy (FIGS) [21,22]. In 2018, IITA developed a FIGS population of drought- and heat-tolerant cowpea. This significantly increased the requests to over 5000 samples distributed to 19 recipients’ countries for research purposes (T. Marimagne, pers.com)

The supply of PGRFA from the genebanks also depends upon the ready availability of a sufficient stock of pest and disease-free materials with legal certainty concerning the conditions under which the materials can be provided and received. Costs associated with multiplying, assuring plant health, and distributing samples of crops that are clonally propagated or have recalcitrant seeds are much higher than for crops of orthodox seed behavior. Under the framework of the CGIAR Genebank CRP (2012–2016) and the CGIAR Genebank Platform (2017–2021), the Crop Trust, in cooperation with the Centers, has developed performance targets and a monitoring system to assess the availability, safety duplication, documentation, and quality management of collections. Ultimately, the target is to have 90% (of accessions in the collections) immediately available and 90% safety-duplicated at two locations (for seed collections only). As of the end of 2019, 78% of all materials in the CGIAR genebank were immediately available, 60% of the seed collection was secured in safety duplication at two levels, and 77% was duplicated at the Svalbard Global Seed Vault. A total of 72% of the clonal crop collection was safety-duplicated in the form of cryopreservation or in vitro cultures in at least one location.

The COVID-19 pandemic has highlighted the strategic importance of cryopreserving clonal crops. In vitro cultures require continuous monitoring and upkeep by genebank staff in personam. If scientists’ access to these collections is limited as a result of governmental policies restricting movement, some accessions in those collections could deteriorate and be lost. If those accessions are not safety duplicated somewhere else, or if the safety duplications are in the form of in vitro collections that are similarly vulnerable to the same risk, then unique materials and potentially unique varieties no longer found in farmers’ fields could disappear if they are not secured in cryopreservation. Cryopreserved back-up collections of these materials would address this risk.

The CGIAR genebanks report that, on occasion, materials they send, and materials they are meant to receive, are held up for long periods of time due to the implementation of phytosanitary regulations, occasionally to the point where materials die before they arrive at their destination.

The agreements between the CGIAR Centers and the Plant Treaty’s Governing Body (signed in 2006) have created legal certainty concerning the status of the collections and the conditions under which they can be distributed. This is reflected in the fact that almost all transfers of PGRFA from the genebanks are under the Plant Treaty’s SMTA (with the exception of materials sent for service agreements, or restoration or direct use by farmers in cultivation, as per the opinions of the Ad Hoc Technical Advisory Committee on the Multilateral System and SMTA) [23]. Just as the Centers received materials under the SMTA from providers in countries that are not contracting parties, the Centers’ genebanks also distributed PGRFA, using the SMTA, to recipients in thirteen countries that are not Plant Treaty contracting parties (between 2012 and 2019 inclusive).

Despite these benefits to the Centers operating under the Plant Treaty’s framework, the genebanks note with concern that some large seed companies, some universities, and one national agricultural research organization are unwilling to receive materials under the SMTA, which makes it impossible for the Centers to distribute materials from the genebank (and also most of the materials from the breeding programs) to them. Other genebanks noted that their ability to distribute materials from the international collections was being constrained by policies of the country in which they are located.

## 4. Discussion

### 4.1. Targeted Acquisitions within the Context of the Global System

Most of the international collections hosted by the CGIAR Centers originated as working collections to support research and breeding programs of international and national public agricultural research organizations. Their subsequent growth depended partly on taking advantage of opportunities to acquire PGRFA from a wide range of sources in an ad hoc manner, for example, being offered material:Previously assembled by other organizations [1];As a result of interactions with scientists from national programs;As part of the international network of base collections organized by the International Board for Plant Genetic Resources (IBPGR) [24];By the Centers’ own breeders.

The collections grew in opportunistic “fits and starts” without resources or tools to systematically analyze their structure and coverage vis-à-vis what exists in situ, or in other collections around the world. Consequently, it can be a challenge for genebank managers to be certain that newly acquired materials are duplicates of materials they already have, or the same as materials that are conserved and made internationally available by other genebanks, or that they are truly unique. Characterization, documentation, and some cross-referencing are key to resolving this issue, but still duplicates or near duplicates are likely to be abundant within and among genebanks In recent years, the Centers’ genebanks have started to take advantage of modern molecular tools such as genotyping to characterize the genetic structure of the collections and to identify genetic differences among and within accessions, for example, potato [25,26,27], sweet potato [28,29], cassava [30], forage grasses [31,32,33], Mexican wheat landraces [34], and African rice [18] IRRI generated whole genome sequences of three thousand accessions in its collection [35]. While still constrained by resource limitations, molecular-level characterization has unprecedented potential to identify redundancies not only within collections, but also across collections, both inside and outside the CGIAR along with identifying potential misclassifications, introgressions, levels of domestication, genetic origins, and putative hybrids. However, it is important to note that even with these data, it is often difficult to determine an adequate cut-off threshold for calling a material a duplicate or too genetically similar for incorporation into the collections. The cost of generating raw sequence data has dropped precipitously, but the expertise and computing power necessary to analyze the data are still expensive and time-consuming, especially when dealing with whole genomes instead of reduced-representation sequencing approaches for genotyping. There are also other biological complications to molecular data alone for resolving duplication issues. However, the CGIAR Genebank Platform has been piloting training programs with scientists from national agricultural research organizations to use genotypic information to analyze within and among accession genetic diversity for a range of crops.

All CGIAR genebanks now have acquisition policies in place and current processes are based on a more critical assessment of whether new materials add diversity to the collections or respond to a specific need or mandate. This is particularly important for clonal collections where, resources permitting, the CGIAR genebanks can use genotyping to help confirm if a sample of newly acquired material is unique (to the collection) before undertaking expensive procedures to test, clean, and reproduce the materials for introduction into the collection as new accessions [31]. Centers working with clonal crops have also developed tools to use genetic sequence information to test for the presence of viruses and bacteria, overcoming costly delays for Centers acquiring and distributing clonal PGRFA [36,37,38].

Under the framework of the CGIAR Genebank Platform, the Centers and the Crop Trust have developed three methods for analyzing gaps in the coverage of their own collections. First, so-called “diversity trees” have been constructed for 22 crops. The trees are developed using published literature and expert knowledge to categorize the diversity held in each crop gene pool into known variety or genotype groups or wild species, which allows the mapping of accessions into the groups and the quantitative representation (or not) of the gene pool by the collection [32,39,40]. Second, spatial analyses have been undertaken using a method to assess the ecogeographic gaps and coverage of current CGIAR crop collections. The method, which works best for collections with a high percentage of available information on the latitude–longitude of the origin of accessions, looks for relationships between geographic patterns in crop distribution with the genetic structuring, and uses these relationships to build distribution models for crop landraces [41]. Third, a method for trait-based gap analyses focuses on the analysis of the distribution of adaptive priority traits in relation to the environment using machine learning to make predictions; it works best where landraces have been associated with an environment for long enough for their traits to become associated with their environment, and presupposes well-characterized collections. Figure 6 includes a preliminarily indication of the coverage of cultivated gene pools in the CGIAR genebank collections through the use of these tools and illustrates clearly that the cultivated diversity of some crops is well represented while others are considerably less.

In addition to the three strategies listed above, some genebanks are utilizing collection-wide genotyping in order to make key decisions on any new acquisitions. This strategy is only effective for genebanks that have collected genotyping data on their entire ex situ collection with a particular marker system. After which, any new material being considered for acquisition can be genotyped with the same marker system (GBS, SNP, DArTseq, etc.) to aid in decision making. The resulting fingerprints of material being considered for acquisition are subsequently compared to the entire germplasm collection to gain genetic insights. Phylogenetic results and genetic distance measures produced from the genotyping data can clearly show which samples are unique and which are redundant to the existing germplasm collection. Once these data are available, decisions can be made on which material to introduce into the genebank, usually under the framework of maximizing genetic diversity and not introducing material that is genetically similar. This is especially a useful strategy in clonal genebanks in which introduction and virus cleaning are expensive and time-consuming.

The Genebank Platform has recently initiated communications with national agricultural research organizations and national Plant Treaty Focal Points in 20 countries to use combinations of these tools as part of an effort to identify complementary holdings and gaps in collections, to identify priority areas for collecting (with the understanding that collected materials would be made available through the Plant Treaty’s multilateral system). At the same time, they undertook to work together to identify potentially useful materials from the international collections to test in the countries concerned. While COVID-19 has had an effect on the preparations for this work, the ambition remains to support the collecting of PGRFA in target countries where there are significant gaps in ex situ conservation.

The CGIAR Centers do not have the objective to host complete collections of the diversity of their mandate crops. Rather, they seek to coordinate efforts with other collections to ensure the broadest possible coverage, long-term conservation, and availability of materials, in the global system as a whole. The Centers will pursue this approach, and the use of the tools and methods described above, in the process of defining collecting priorities and revising the current set of crop conservation strategies (available online at https://www.croptrust.org/resources/#ex-situ-conservation-strategies). The Centers (and the Crop Trust) do not organize collecting expeditions on their own. They work in collaboration with NARS partners who take responsibility for conducting the collecting, obtaining requisite permissions in compliance with national laws and regulations, and first depositing samples in their own national genebanks before transferring them to the CGIAR genebanks.

### 4.2. Phytosanitary Standards and Initiatives

Maintaining the health of the germplasm in the international collections is critically important to ensure against the international spread of quarantine pests and diseases. In 2019, the 11 germplasm health units (GHUs) of the 10 Centers hosting PGRFA collections collectively facilitated 2004 exchanges of materials with 141 countries. Altogether 152,469 samples of 105,961 accessions were analyzed by performing 594,909 diagnostics tests, and 13,248 samples were rejected to be replaced with healthy materials from other batches or to be subject to phytosanitary treatments to eliminate the pests concerned [40]. Most of the pests detected are target-specific species. Viral pathogens are most frequently intercepted in legume seeds and in clonal crops. Centers only distribute germplasm that is free of quarantine organisms. Untested and unclean materials are not distributed until establishment of the pest-free stocks. As of 2019, nearly 80% of the germplasm conserved in CGIAR genebanks had been tested for quarantine pests and about 78% of the collection was available for immediate distribution.

The International Plant Protection Convention (IPPC) is essential for promoting and harmonizing countries’ phytosanitary regulations, and for facilitating international movement of plant genetic resources that are free of quarantine pests and diseases. However, the system still can and should be further tailored to address the particularities of the international movement and uses of plant genetic resources. Under the framework of the IPPC, contracting parties occasionally adopt International Standards for Phytosanitary Measures (ISPMs) that are tailored for different subject matters that could be a potential carrier of quarantine pests and diseases over international borders (e.g., farm equipment, commercial seed). To date, approximately 42 ISPMs have been adopted. Unfortunately, no ISPM for the regulation of international movement of plant germplasm from genebanks has been developed. The ISPM-36 on Integrated Measures for Plants for Planting and the ISPM-38 on International Movement of Seeds partially address some of the issues, but insufficiently. As a result, the national plant quarantine facilities of many countries either develop and follow their own norms or follow those prescribed through ISPMs dealing with commercial seed (or plantlets), which are not appropriate for plant germplasm shipped to and from genebanks. The commercial seed ISPMs—which anticipate tons of seed in a single shipment—prescribe testing protocols that deplete most of the very small quantities of seed transferred to and from genebanks, or create long delays which are unnecessary to address risks associated with genebank germplasm. These delays can result in materials dying in transit before they arrive at their intended destinations, or they arrive so late that they miss an entire planting season, thereby contributing to delays or cessation of planned research and or plant breeding activities. While phytosanitary issues are most challenging for recalcitrant seed or clonal crops, the genebanks report similar challenges with seed crops.

To address this situation, CGIAR germplasm health experts are working with partners from other organizations to develop a draft protocol for a comprehensive phytosanitary compliance assurance procedure that will demonstrate the best procedures in use for germplasm production and health assurance, while maintaining transparency in risk assessment and mitigation strategies to get NPPO accreditation as trusted to fast-track germplasm distribution. The initiative is referred to as the CGIAR GreenPass Phytosanitary Protocol (GreenPass). If the concept is endorsed at the level of the IPPC’s Commission on Phytosanitary Measures, NPPOs could justify eliminating redundant checks by cutting some steps or reducing the processing time for material from GreenPass-accredited facilities. (These issues are addressed in more detail by Kumar et al. in this Special Edition of *Plants*.)

### 4.3. Genebank and Plant Genetic Resources Valuation

During the late 1990s, the genebanks received mounting criticism that materials stored in genebanks were rarely used. This was a concern because necessary investments in genebanks are difficult to obtain if potential investors do not appreciate the value of crop diversity, including the multitude of services and benefits it can provide. Several studies followed in the early 2000s which contradicted the viewpoint that genebanks were “unused” [42,43] and a large body of research documented the high rates of return from the genetic improvement of crops for yield, yield stability, quality, nutritional composition, resource use efficiency, and resistance to pests and diseases [44,45,46,47,48,49]. Most of the economic benefits have been generated from farm productivity gains which can be attributed to research and breeding programs by publicly funded institutions, such as the CGIAR, and society and consumers have especially benefited from lower food prices. Since then, however, the research on the value of international collections and the genetic diversity held in the CGIAR genebanks has not kept pace.

Recently, the CGIAR Genebank Platform and the Crop Trust supported the establishment of a Community of Practice on Genebank Impacts to revive the interest in applied economics research in this area. The work resulted in several papers which have made important contributions to earlier research on genebank valuation [50]. For example, Bernal-Galeano et al. (2020) [51] estimated a gross benefit from the “Victoria” potato variety in Uganda at USD 1.04 billion dollars, which exceeds the annual operational cost of the CIP genebank several times over. Villanueva et al. (2020) [52] estimated that a 10% increase in the genetic contribution of IRRI genebank accessions to an improved rice variety grown by rice farmers in East India is associated with a yield increase of 27%. Aberkane et al. (2020) [53], Sellitti et al. (2020) [54], and Kitonga et al. (2020) [55] focused on the use of wild, semi-natural, and landrace genetic materials in enhancing crop diversity options for breeders and farmers. The study by Ocampo-Giraldo et al. (2020) [56] highlighted the importance of combining ex situ and in situ approaches in a dynamic model of conservation. Alexandra et al. (2020) [57] narrated the formation of Pacific Community’s Centre for Pacific Crops and Trees (CePaCT) in Fiji and underscored the role of a global effort to collect, conserve, and breed taro in response to disease outbreaks.

This current set of studies attests to the value of genebanks in at least two ways. First, they contribute to a better understanding of the role, function, and value of genebanks, in the light of climate change and evolving food security challenges. Several authors were able to trace the ancestry of varieties currently adopted by farmers to specific accessions in the genebanks and apportion economic gains by drawing from information on pedigrees. Second, the studies highlight the importance of long-term conservation and safety duplication of unique and diverse genetic materials, for the potential unknown use of future generations. As with Gollin (2020) [58], the findings support the need to refocus global conservation strategies on the efficient management of genetic resources, including on acquisition and conservation priorities.

### 4.4. Money, Politics, and Law

The findings presented above concerning the influence of political tensions, restrictive policies, and legal uncertainty are similar to the conclusion of the previous study [1] of factors affecting Centers’ acquisitions between 1980 and 2009.

On the positive side, there is evidence that the Plant Treaty’s multilateral system of access and benefit sharing is contributing positively to the willingness of many countries, national genebanks, and other providers to make PGRFA available and to safety-duplicate material in the CGIAR Center-hosted international collections. Perhaps the most significant evidence (which is so obvious it is often overlooked) is that all of the material in the international collections ultimately came from countries and, to date, 146 countries and the EU have voluntarily ratified the Plant Treaty, which invites the Centers to manage those collections under the Plant Treaty framework, and to make those materials available under the SMTA. As of November 2019, at least 135,000 accessions maintained in CGIAR genebanks (approximately 17% of the collections) were originally obtained from countries that were not Plant Treaty contracting parties including, but not limited to, the following countries: Mexico, China, Nigeria, Colombia, Thailand, Russia, Vietnam, Azerbaijan, Republic of South Africa, Uzbekistan, and Kazakhstan. It is partly for this reason that the CGIAR Centers feel duty bound to continue making materials available under the SMTA to recipients in countries that are not Plant Treaty members. Some contracting parties, particularly those who have been criticized in recent years for not making more PGRFA directly available through the multilateral system, complain that there is insufficient recognition of the fact that much of the genetic resources that the CGIAR Centers’ genebanks distribute through the multilateral system (with the exception of accessioned breeders materials) originally came from them. Information about the country sources of materials (provenance) held in the Center-hosted international collections is available through Genesys, and Centers have published papers highlighting the origins of materials in the collection [31,53]. However, more can and should be done—through more popular, less expert-oriented mechanisms—to celebrate countries’ contributions to the international collections and the multilateral system. Indeed, during the Plant Treaty Governing Body meeting in October 2019, CGIAR undertook to work with the Plant Treaty Secretariat to publicize this information more broadly.

More recent evidence of the positive influence of the Plant Treaty on providers’ willingness to include materials in the international collections is the surge of materials received by the Centers’ genebanks between 2010 and 2019. All of those materials were either provided under the SMTA or with permission for Centers to subsequently make those materials available under the SMTA. It seems likely that many of those providers would not have been willing (or permitted) to provide materials for inclusion in the international collections in the absence of the Plant Treaty, the multilateral system, and the SMTA. Indeed, some genebanks have been informed by national partners that while those partners can provide PGRFA of Annex 1 materials, they cannot provide PGRFA of non-Annex 1 materials, because their national rules only apply to materials formally included within the scope of the multilateral system.

The fact that the Centers were able to acquire materials—particularly materials through new collecting expeditions—is evidence that other regulatory frameworks, apart from the Plant Treaty’s multilateral system, are also contributing to the willingness/ability of providers to provide access to those materials. A lot of in situ PGRFA are not automatically included in the multilateral system, so collection must be approved by a competent authority subject to national law (or in the absence of a law, some other standard observed by the authority) to allow the material to be collected, deposited in the national genebank, and later sent under the SMTA to the CGIAR genebanks. (In this context, it is relevant to note that some countries have explicitly adopted the policy of not wanting to regulate access in this manner). The CGIAR genebanks never collect on their own. They work in partnership with national organizations who manage the interface with national authorities, logistics, and the actual collecting. Transfer within a country of material from one ABS system to another requires coordination and cooperation between the competent authorities and stakeholders.

It is certainly the case that the existence of the SMTA (because it is standard) made it possible for the Centers to process agreements to receive and manage those same materials. It would have been impossible for the Centers to negotiate unique transfer agreements with sui generis benefit sharing, dispute resolution, scope of use, and other conditions in each case, and then to put systems in place to manage materials under a plethora of different conditions.

It is important to consider the significance of the fact that most of the materials made available to the genebanks between 2010 and 2019 were associated with the Regeneration and CWR projects. First, it highlights the fact that conserving and providing PGRFA require financial and technical support and coordination, and in the absence of such support, many potential actors in the global system are unable to play their anticipated roles including securing the unique materials in their collections through safety duplication. In this context, it is important to emphasize that the funds and/or other non-monetary benefits provided to project partners in the Regeneration and CWR projects were modest, designed to subsidize/cover the costs of regeneration and or collecting, health testing and shipping, and in some cases training to do those things and for the Centers to receive the materials. Extra funding is needed to support collecting new PGRFA. Those costs, transaction costs in particular, would decrease if countries had well-defined systems for processing requests for germplasm, and clear signals from competent authorities that such requests were acceptable to consider. It is interesting to note that the USDA has a dedicated Plant Exploration Program managed by a Plant Exchange Office that is responsible for organizing collecting expeditions, reaching out to national competent authorities on behalf of the genebanks included within the national system [3]. There is no such service or function within the CGIAR. Second, the Regeneration and CWR projects highlight the continued importance, and need for, scientifically informed priority setting and international coordination to generate a shared sense of purpose and to motivate the wide range of actors, spread across the world, who ended up being engaged in those projects. The rounds of projects supported by the Plant Treaty’s Benefit-sharing Fund (BSF) have played a similar role, catalyzing interest and engagement of a broad range of actors operating from the levels of individual farms to international organization, and providing financial and technical support for project participants to, among other things, collect, multiply, health-test, and share PGRFA through the multilateral system. It would be interesting, but it is beyond the scope of this paper, to identify the range of materials included in the multilateral system in general as a result of the BSF projects.

This is not to say that all of these internationally coordinated projects worked perfectly, or were immune from criticisms, or that many contracting parties, organizations, and people involved in the conservation and use of PGRFA do not have alternative visions or priorities. The point is that while the Plant Treaty’s multilateral system of access and benefit sharing provides an internationally sanctioned, clearly defined legal platform of exchanging plant genetic resources, it is clear that additional financial and technical support, and inspiring visions for dispersed actors to work together, are also necessary to take advantage of the multilateral system. Other actors (including potentially the Plant Treaty’s own Governing Body, or, more realistically, a specialist group it establishes) could develop a vision and internationally coordinated program that similarly stimulate and support activities that take advantage of the multilateral system to access, improve, and share PGRFA, and safeguard threatened PGRFA in furtherance of Sustainable Development Goals (e.g., pooling and evaluating a genetically diverse range of PGRFA for traits adapted to local climate changes).

Meanwhile, the CGIAR Centers’ genebanks continue to distribute hundreds of thousands of PGRFA samples under the Plant Treaty’s SMTA. Indeed, in the decade between 2010 and 2019 (inclusive), the Centers’ genebanks distributed 21% more PGRFA samples than they did over the previous decade. These numbers attest to the utility of the multilateral system and increased reliance upon it as a means of accessing genetic materials.

The Plant Treaty’s positive impact on the availability of PGRFA more generally (i.e., without involvement of the Center’s genebanks) is further evidenced by the fact that, since 2014, sixteen additional countries became members of the Plant Treaty. Among them, is the USA, with the result that approximately 500,000 additional PGRFA accessions are available through the multilateral system of access and benefit sharing. In addition, during the same period, one additional international organization, the International Center for Biosaline Agriculture (ICBA), agreed to make PGRFA available under the terms of and conditions of the Plant Treaty.

On the negative side, however, there is also evidence that a number of Plant Treaty contracting parties continue to be reluctant to implement the multilateral system. Overall, national level implementation of the multilateral system is still relatively low. This is perhaps most clearly manifested in the fact that only 44 out of 146 countries that are contracting parties have confirmed what PGRFA within their borders are actually included in the multilateral system. (see “Material Available in the Multilateral System” on the Plant Treaty website at http://www.fao.org/plant-treaty/areas-of-work/the-multilateral-system/collections/en/). This may not be a “black letter law” obligation under the Treaty framework, but it is a commonly acknowledged prerequisite for the multilateral system to practically function [59]. Furthermore, some of the countries that have provided notice have confirmed only a fraction of the collections maintained in their national genebanks or other national public organizations. There are even fewer notices to the Governing Body concerning PGRFA that are voluntarily included by natural and legal persons. Few countries have reported to the Governing Body that they have adopted new policies or administrative measures to implement the multilateral system. On one hand, it is not required by the Plant Treaty that a country develop new policies or laws to implement the multilateral system; on the other hand, many countries report that in the absence of new policy instruments approved by the national government that clarify who has the right to provide materials under the multilateral system, they are unable to do so [59]. In such cases, the absence of new policy measures does indeed reflect a lack of national level implementation.

The relatively slow rate of national level implementation by a number of countries can be partially accounted for by the fact that developing country contracting parties are dissatisfied by the fact that, to date, there has been only one payment to the Plant Treaty’s Benefit-Sharing Fund by a commercial user of materials from the multilateral system. This dissatisfaction contributed to the Fifth Session of the Governing Body in 2013 launching a process to enhance the functioning of the multilateral system. That process continued until 2019, when it was suspended, with no new agreement reached, and high levels of unresolved political tension between contracting parties.

International tension and disagreement concerning access and benefit-sharing issues in other fora have also spilled over into meetings under the Plant Treaty framework. The Nagoya Protocol came into force in 2010. It was designed to address (mostly developing) countries’ concerns that the Convention on Biological Diversity (CBD) did not sufficiently promote its benefit-sharing objective. However, by 2013, the Conference of the Parties to the CBD in Mexico was overtaken by tensions over benefit sharing from commercial use of DSI, which is not addressed by the Nagoya Protocol (or at least not in a way that the international community agrees upon). Since then, the issue has dominated the agendas of the CBD, Plant Treaty, FAO CGRFA, and others. Under these circumstances, some parties who are not content with levels of monetary benefit sharing from the use of PGRFA and/or genomic sequence information may be reluctant to make more materials available through the multilateral system by depositing them in the international collections hosted by the CGIAR Centers until there is some resolution to the DSI and ABS issue.

The previous study [1] of factors affecting the Centers’ germplasm acquisitions from 1980 to 2009 adopted an implicitly millennial framework, looking forward to a new era when international tensions over access and benefit sharing were resolved, and national laws implementing international agreements provided requisite legal certainty for PGRFA providers and recipients alike. In 2010, it seemed reasonable to expect that this state of affairs could be achieved within a few years. Since that time, tensions over access and benefit sharing have increased, and the possibility of arriving at a set of final international agreement(s) on the issue appears to have receded still further into the distance.

## 5. Conclusions

Over the course of the decade 2010–2019, the CGIAR Center genebanks received a surge of PGRFA from providers around the world, with permission to make those materials available through the Plant Treaty’s multilateral system of access and benefit sharing. Most of those newly deposited materials were associated with an internationally coordinated project that provided financial and technical support for the providers to regenerate and safety-duplicate unique, at-risk PGRFA in ex situ collections around the world.

The Regeneration and CWR projects, the BSF project cycles, and the CGIAR genebanks’ experiences over the last ten years highlight the critical importance of internationally coordinated projects to motivate and instill otherwise dispersed and disconnected actors with a sufficiently shared sense of purpose, to make materials available through the multilateral system. Those projects and experiences also underscore the importance of providing modest levels of financial support to cover both the providers’ and the recipient genebanks’ costs associated with collecting, multiplying, health-testing, sending, and receiving PGRFA, and of providing technical support/training to providers/partners.

The number of collecting expeditions organized by CGIAR Centers in this period was higher than in the previous decade, but lower than some national genebanks which play an important role in the global system of conservation, sustainable use, and exchange of plant genetic resources. Increasingly sophisticated and globally coordinated gap analyses are making it possible to identify where gaps in collections exist. Gap analyses presented in this paper highlight that ex situ collections managed by the CGIAR represent well the cultivated gene pool of some crops, while there are significant gaps for other crops. Given the need to keep costs to a minimum, and the possibility of sharing responsibilities with other global system actors, CGIAR Centers must redouble their efforts to address the results of gap analyses in concert with other organizations that host internationally available collections.

While financial support, scientific leadership, and international coordination are indispensable, so too are supportive policies. The CGIAR genebanks’ experiences highlight the importance of supportive policies, and conversely, the negative impacts of restrictive or unclear policies and laws, on their ability to acquire new materials to include in the international collections, and to distribute those materials to recipients around the world. In retrospect, it is perhaps surprising that over the same period of time, the international community launched and attempted to renegotiate the multilateral system of access and benefit sharing—with all the uncertainty that attends an international negotiation—and the Centers’ genebanks enjoyed a surge of new materials being made available to include in the international collections. During the same period of time, they continued to distribute an extraordinary diversity of PGRFA to recipients around the world (even more than in the previous decade), a fact which reflects the persistent need/demand for access to those materials for agricultural research and development, the deep rooted nature of the CGIAR collections within the global system, and the positive influence of the Centers’ Article 15 agreements with the Governing Body of the Plant Treaty.

However, with the suspension of the process to enhance the multilateral system, and widespread tensions concerning the governance of digital genomic sequence information, international disagreement over access and benefit sharing is becoming still more geopolitically polarized. There is a significant risk that this increased polarization could further undermine the willingness of a range of actors to make materials available through the multilateral system in general, and to the CGIAR genebanks in particular. The Centers highlight the importance of resolving those tensions to “head off” unintended potential negative impacts on the CGIAR’s mission, the global system, and the SDGs. Rapid loss of biodiversity, climate change, the COVID-19 pandemic, rising populations, depleted soils, and a range of other challenges make the conservation, availability, and use of PGRFA more important than ever. It is essential that the Plant Treaty (and Nagoya Protocol and IPPC) is implemented in ways that support all actors in the global system to fulfill their roles.

## Figures and Tables

**Figure 1 plants-09-01296-f001:**
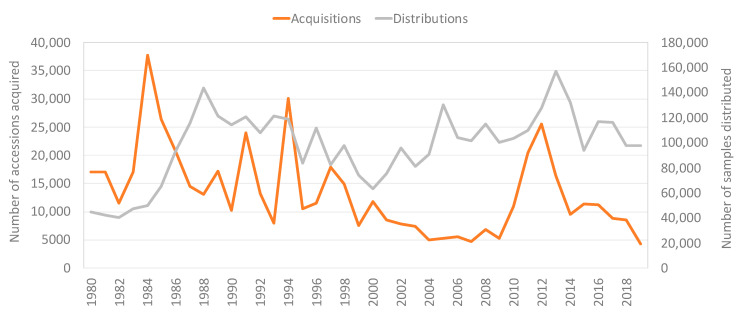
Acquisitions and distributions by all CGIAR Centers 1980–2019.

**Figure 2 plants-09-01296-f002:**
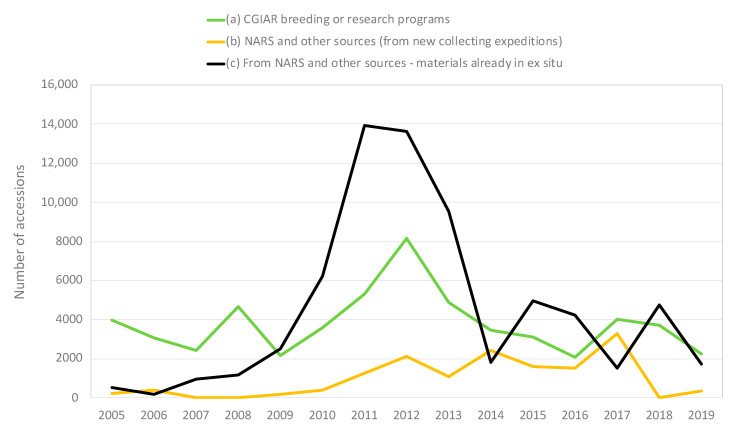
Details of CGIAR genebank acquisitions 2005–2019.

**Figure 3 plants-09-01296-f003:**
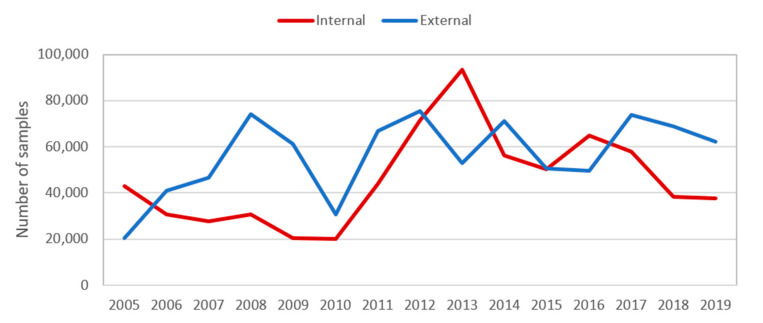
Centers’ distributions, 2010–2019, broken down by (i) transfers within the CGIAR (internal), and (ii) transfers to recipients outside the CGIAR (external).

**Figure 4 plants-09-01296-f004:**
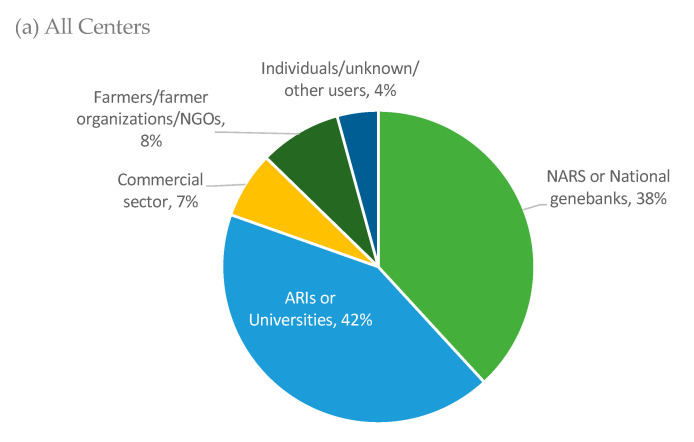
Recipients of germplasm distributed by CGIAR genebanks, 2017–2019.

**Figure 5 plants-09-01296-f005:**
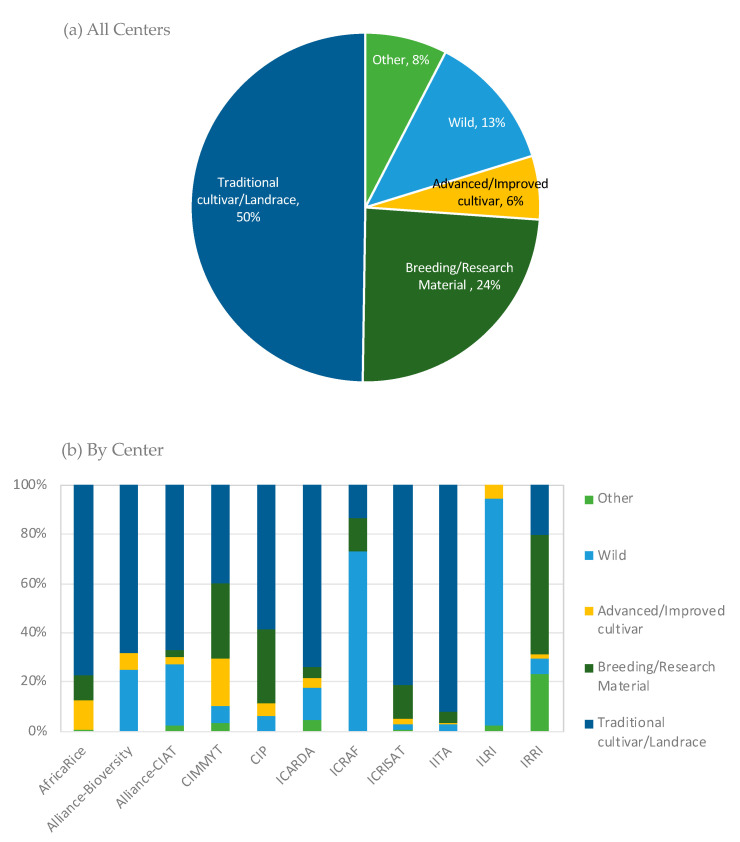
Types of germplasm distributed by CGIAR genebanks, 2017–2019.

**Figure 6 plants-09-01296-f006:**
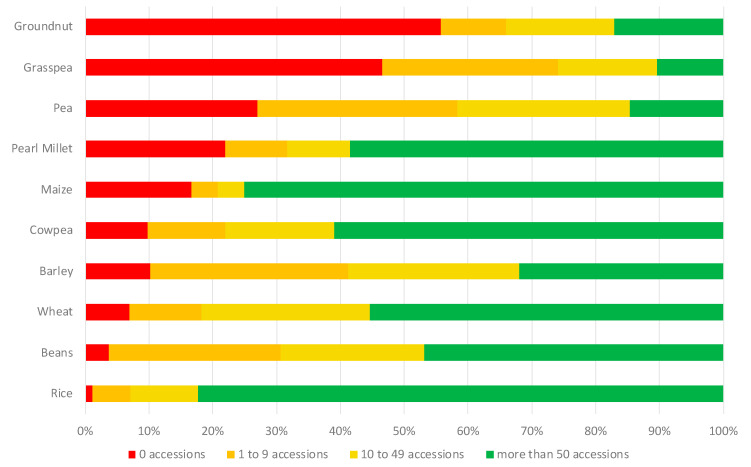
Preliminary assessment of the coverage of traditional landraces in international collections managed by CGIAR Centers. Note: these findings were generated by mapping accessions onto diversity trees. The colors correspond to numbers of accessions representing identified landrace end-groups or varieties making up the crop gene pool. This figure is reproduced from the 2019 Annual Report of the CGIAR Genebank Platform [40].

**Table 1 plants-09-01296-t001:** Providers of materials acquired by CGIAR genebanks 2010–2019 (excluding CGIAR breeding programs)—Notes: associated with Regeneration project (+), CWR project (*). We separate the two genebanks hosted by the Alliance of Bioversity and CIAT.

CGIAR Genebanks (Crop Mandate)	From New Collecting Mission	From Existing Ex Situ Source	
**AfricaRice**			
(rice)	Benin	Burkina Faso +	
	Burundi	Guinea +	
	Cameroon	Kenya	
	DR Congo	Mali +	
	Gambia	Myanmar	
	Senegal		
	Tanzania		
	Uganda		
**Alliance-Bioversity**			
(banana)	Cook Islands	Cameroon +	Nigeria
	Indonesia	China	Philippines +
	Samoa	DR Congo	PNG +
		Germany	Thailand
		India +	Uganda +
		Indonesia +	USA
		Japan	Vietnam +
		Myanmar	
**Alliance-CIAT**			
(beans, cassava,	Costa Rica *	Azerbaijan +	Myanmar +
forages)	El Salvador *	Colombia	Nicaragua +
		Costa Rica +	Peru
		Dominican Republic +	Puerto Rico +
		El Salvador +	UK
		Malaysia	USA
**ICARDA**			
(wheat, barley,	Armenia *	Afghanistan	Kyrgyzstan
forages, grasspea,	Cyprus *	Albania	Latvia
pea, lentils,	Georgia	Algeria	Lebanon *
chickpea,	Greece	Argentina	Libya
faba bean)	Jordan	Armenia *	Lithuania
	Kazakhstan	Australia	Macedonia
	Lebanon *	Austria	Mexico
	Russian Federation	Azerbaijan *	Moldova
	Tajikistan	Belarus	Mongolia +
		Bhutan	Montenegro
		Bolivia	Morocco
		Bosnia and Herzegovina	Nepal
		Brazil	Netherlands
		Bulgaria	New Zealand
		Canada	Pakistan
		Chile	Palestinian Authority
		China	Peru
		Colombia	Poland
		Croatia	Portugal *
		Cyprus *	Romania
		Czech Republic	Russian Federation
		Czechoslovakia	Saudi Arabia
		Denmark	Serbia
		Ecuador	Slovakia
		Egypt	Spain
		Finland	Sudan
		France	Sweden
		Georgia *	Switzerland
		Germany	Syria
		Greece	Tajikistan
		Hungary	Tunisia
		India	Turkey
		Iran	Turkmenistan
		Iraq	Ukraine
		Italy *	UK
		Japan	Uruguay
		Jordan	USA
		Kazakhstan	Uzbekistan
		Korea, Republic of	Yugoslavia
**CIMMYT**			
(wheat, maize)		Afghanistan	Italy
		Albania +	Kenya
		Australia	Japan
		Austria	Mexico +
		Azerbaijan +	Mongolia +
		Belarus +	Nepal +
		Brazil	Nicaragua +
		Bulgaria +	Nigeria
		Canada	Paraguay
		China	DPR Korea +
		Ecuador +	Peru +
		England	Philippines +
		Ethiopia	S. Africa
		France	Syria
		Georgia +	Tanzania +
		Honduras +	Turkey
		Hungary	UAE
		India	Ukraine +
		Indonesia +	Uruguay +
		Iran	USA
		Ireland	Zambia +
		Israel +	Zimbabwe
**CIP**			
(Andean roots	Peru *	China	Rwanda +
and tubers,		Ecuador +	Uganda +
sweet potato,		Indonesia +	USA
potato)		Peru +	
**ICRAF**			
(trees)		China	Senegal
		Kenya	Tanzania
		Mali	Uganda
**ICRISAT**			
(finger millet,	Burkina Faso	Azerbaijan +	Nepal +
pearl millet,	Ghana	Benin +	Niger +
sorghum,	India	Bulgaria +	Nigeria +
small millets,	Kenya	Burkina Faso +	Senegal +
pigeon pea,	Niger	Burundi +	Sudan +
chickpea)	Nigeria	Canada	Tanzania
	Uganda	China	Togo +
	Zimbabwe	Georgia +	Uganda +
		Ghana +	Uzbekistan +
		India +	Yemen +
		Kenya +	Zambia +
		Mali +	Zimbabwe +
**IITA**			
(yam, maize,	Cameroon	Afghanistan +	Mali +
cowpea,		Australia +	Mexico +
cassava, banana,		Azerbaijan +	Mozambique +
Bambara		Benin +	Namibia +
groundnut,		Botswana +	Niger +
misc. legumes,		Brazil +	Nigeria +
cocoyam)		Burkina Faso +	Norway
		Burundi +	Oman +
		Cape Verde +	Paraguay +
		Central African Republic +	Philippines +
		Chad +	Puerto Rico +
		Chile +	Russia +
		China +	Rwanda
		Colombia +	Senegal +
		Costa Rica +	Sierra Leone +
		Cote d’Ivoire +	Somalia +
		DR Congo +	South Africa +
		Egypt +	Sudan +
		France	Suriname +
		Gabon +	Swaziland +
		Georgia +	Taiwan +
		Ghana +	Tanzania +
		Guatemala +	Thailand +
		Guinea +	Togo +
		Hungary +	Turkey +
		India +	Uganda +
		Indonesia +	UK +
		Iran +	USA +
		Italy +	Vietnam +
		Kenya +	Yemen +
		Lesotho +	Zambia +
		Madagascar +	Zimbabwe +
		Malawi +	
**ILRI**			
(forages and fodder)		Brazil	USA
**IRRI**			
(rice)		Bangladesh	Malaysia +
		Brazil	Myanmar +
		Cambodia	Nepal +
		China	Pakistan +
		DPR Korea +	Philippines +
		France	Tanzania
		India	Turkey
		Indonesia +	Uganda
		Iran	UK
		Lao PDR +	USA
		Madagascar +	Vietnam +

**Table 2 plants-09-01296-t002:** 20 countries which received the most samples from the CGIAR genebanks, 2017–2019 (not including intra- and inter-CGIAR Center transfers).

Country	2017	2018	2019	Total
1 India	11,439	15,154	5693	32,286
2 China	6480	4311	3846	14,637
3 United States	3844	3727	1543	9114
4 Italy	6834	424	1767	9025
5 Mexico	1818	3701	2667	8186
6 Morocco	2736	3780	1016	7532
7 Nigeria	2187	2471	2069	6727
8 Ethiopia	1587	1134	1792	4513
9 Australia	3600	168	611	4379
10 Colombia	519	1117	2338	3974
11 Peru	1109	634	2049	3792
12 Mali	1577	1679	459	3715
13 Japan	949	1874	409	3232
14 Sudan	62	34	3116	3212
15 Germany	2124	1011	62	3197
16 Belgium	184	2742	78	3004
17 Lebanon	-	2071	839	2910
18 Kenya	378	984	1411	2773
19 United Kingdom	429	1839	322	2590
20 Iran	643	100	1299	2042

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
