# Peer review of "Germplasm Acquisition and Distribution by CGIAR Genebanks"

_plants, 2020, doi:10.3390/plants9101296_

Round 1
Reviewer 1 Report
This article presents all aspects of CGIAR long-term work, including the collection, development and application of methods for collecting, storing, studying, documenting and distributing the world-wide plant genetic resources in the conditions of complex natural and geopolitical challenges. Such type of articles should be published on a regular basis. I recommend the article for publication.
Author Response
No changes were necessary in response to this review
Reviewer 2 Report
I think the manuscript is far too long and does not keep to the rather focused / specific title. It could almost stop at the end of the Findings section.
Otherwise, I have made edits / suggestions to the attached. Please note I had already downloaded and started reviewing the version that was already submitted, and this is what I have commented on.

Author Response
The reviewer made numerous comments and suggested small revisions and edits throughout the entire text. We have systematically reviewed them all, making changes to the paper throughout. All of these changes are in track changes mode; they are far too numerous to describe in a separate document.
We did not delete the discussion and conclusion sections of the document which this reviewer raises as a possibility. The other two reviewers did not make such suggestions.
Reviewer 3 Report
Please see attached.

Author Response
We found this reviewer's comments to be the most useful. We revised the introduction to clearly articulate the research questions the paper addresses. We carefully reviewed that the findings, discussion and conclusion address, and respond to the questions as articulated. We appreciate the stimulous from the external reviewer to go through this exercise.